# USP5 inhibits anti-RNA viral innate immunity by deconjugating K48-linked unanchored and K63-linked anchored ubiquitin on IRF3

**Zigang Qiao[1,2]ᵒ, Dapei Li[1]ᵒ, Fan Zhang[1]ᵒ, Jingfei Zhu[1], Siying Liu[1], Xue Bai[1,2], Haiping Yao[1], Zhengrong Chen[3], Yongdong Yan[3], Xiulong Xu[2]\*, Feng Ma[1]\***

**1** National Key Laboratory of Immunity and Inflammation, and CAMS Key Laboratory of Synthetic Biology Regulatory Elements, Suzhou Institute of Systems Medicine, Chinese Academy of Medical Sciences & Peking Union Medical College, Suzhou, China, **2** Institute of Comparative Medicine, College of Veterinary Medicine, Yangzhou University, Yangzhou, China, **3** Department of Respiratory Medicine, Children's Hospital of Soochow University, Suzhou, China

ᵒ These authors contributed equally to this work.

\* xxl@yzu.edu.cn (XX); maf@ism.pumc.edu.cn (FM)

**Data Availability Statement:** All relevant data are within the manuscript and its Supporting Information files. The microarray analysis raw data

## Abstract

Interferon regulatory factor 3 (IRF3) is a central hub transcription factor that controls host antiviral innate immunity. The expression and function of IRF3 are tightly regulated by the post-translational modifications. However, it is unknown whether unanchored ubiquitination and deubiquitination of IRF3 involve modulating antiviral innate immunity against RNA viruses. Here, we find that USP5, a deubiquitinase (DUB) regulating unanchored polyubiquitin, is downregulated during host anti-RNA viral innate immunity in a type I interferon (IFN-I) receptor (IFNAR)-dependent manner. USP5 is further identified to inhibit IRF3-triggered antiviral immune responses through its DUB enzyme activity. K48-linked unanchored ubiquitin promotes IRF3-driven transcription of IFN-β and induction of IFN-stimulated genes (ISGs) in a dose-dependent manner. USP5 simultaneously removes both K48-linked unanchored and K63-linked anchored polyubiquitin chains on IRF3. Our study not only provides evidence that unanchored ubiquitin regulates anti-RNA viral innate immunity but also proposes a novel mechanism for DUB-controlled IRF3 activation, suggesting that USP5 is a potential target for the treatment of RNA viral infectious diseases.

## Author summary

Innate immunity is the first line of defense against highly pathogenic RNA viruses. IRF3 plays a crucial role in initiating IFN-I-dependent antiviral innate immunity. Post-translational modifications, particularly ubiquitination, control the stability and transcription activity of IRF3. However, it is unknown whether unanchored ubiquitination and deubiquitination of IRF3 involve modulating antiviral innate immunity against RNA viruses. Here, we show that IFN-I-mediated USP5 suppression enhances K48-linked unanchored and K63-linked anchored ubiquitination of IRF3, thereby promoting IRF3 activation and further IFN-β transcription. We demonstrate a protective role of the IFN-I-mediated

(accession no. GSE49840) were downloaded from GEO.

**Funding:** This work was supported by the National Key Research and Development Program of China (2018YFA0900803 to FM), the National Natural Science Foundation of China (32400726 to FZ, 82301982 to JZ, 32270924 to DL, and 32170880 to FM), the Natural Science Foundation of Jiangsu Province (BK20230281 to JZ, BK20221256 to DL, and BK20200004 to FM), Non-profit Central Research Institute Fund of CAMS (2019PT310028 to FM), the Special Research Fund for Central Universities, Peking Union Medical College (3332024090 to FZ), CAMS Innovation Fund for Medical Sciences (2021-I2M-1-041, 2021-I2M-1-047, 2022-I2M-2-004, and 2022-I2M-2-010 to FM), The Suzhou Municipal Key Laboratory (SZS2023005 to FM), NCTIB Fund for R&D Platform for Cell and Gene Therapy, and 333 High-level Talent Training Project to FM. The funders had no role in study design, data collection and analysis, decision to publish, or preparation of the manuscript.

**Competing interests:** The authors have declared that no competing interests exist.

downregulation of USP5 during host infected with RNA viruses. Our study also shows the dual cleavage activity of USP5 in deconjugating K48-linked unanchored and K63-linked anchored polyubiquitin on IRF3, which provides insights into the molecular mechanisms governing host defense against RNA virus infection.

## Highlights

- USP5 is downregulated during host anti-RNA viral innate immunity in an IFNAR-dependent manner
- USP5 inhibits antiviral innate immunity against RNA viruses
- Unanchored K48-linked polyubiquitin chains promote IRF3 activation
- USP5 deconjugates both K48-linked unanchored and K63-linked anchored polyubiquitin on IRF3

## Introduction

In recent years, the RNA viruses have caused newly emerging and reemerging infectious diseases that present a significant risk to human health and survival [1]. Following infection by RNA viruses, the presence of foreign viral nucleic acids is detected by RIG-I-like receptors (RLRs), such as retinoic acid-inducible gene I (RIG-I) and melanoma differentiation associated gene 5 (MDA5). Upon binding to viral RNAs, the CARD domains of RIG-I or MDA5 are exposed [2–5], facilitating the interaction with the downstream adapter protein mitochondrial antiviral signaling protein (MAVS, also known as VISA, CARDIF, or IPS-1) [6–9]. Activated MAVS then recruits two cytosolic protein kinase complexes, TANK-binding kinase 1 (TBK1) and inducible I-kappaB kinase (IKKi) [10], which in turn activate the transcription factors interferon regulatory factor 3/7 (IRF3/7) [11], thereby inducing the expression of type I interferon (IFN-I) to establish the antiviral state in the infected host [12]. Effective clearance of invaded viruses requires sufficient activation of IRF3. However, proper termination of IRF3 activation is crucial to prevent immunopathology, including type I interferonopathies such as systemic lupus erythematosus [13]. To efficiently eliminate invaded pathogens while avoiding undesirable damage, a comprehensive understanding of the molecular mechanisms regulating IRF3 activation becomes extremely important.

The expression and activation of IRF3 are strictly regulated by ubiquitination and deubiquitination. The K48-linked ubiquitination is commonly associated with IRF3 proteasomal degradation [14]. Several E3 ligases, including MID1 [15], UBE3C [16], RNF5 [17], TRIM26 [18], and c-Cbl [19], have been reported to enhance the K48-linked ubiquitination and proteasomal degradation of IRF3 to prevent excessive immune responses. A recent study indicates that the deubiquitinase BAP1 targets IRF3 to govern its stability by removing K48-linked ubiquitin chains during the innate immune responses [16]. In addition, the K63-linked ubiquitination has been identified as an IRF3 activator during viral infection [20]. A functionally deficient form of OTUD1 leads to the over-activation of innate immune signaling by maintaining the K63-linked ubiquitination of IRF3 [21]. However, the mechanisms regulating the turnover of K63-linked ubiquitination of IRF3 in antiviral immunity are not yet fully understood.

DUBs deconjugate anchored ubiquitin chains by breaking the isopeptide bond between the C-terminal glycine of ubiquitin (G76) and the lysine of the substrate, or between adjacent ubiquitin molecules within polyubiquitin chains [22,23]. Recent studies have demonstrated that unanchored polyubiquitination plays a key role in controlling innate immunity. The K63-linked unanchored polyubiquitin chains directly activate TAK1 and IKK [24], and the K63-linked unanchored polyubiquitination of RIG-I and MDA5 has been shown to lead to the activation of the IFN-I signaling pathway [25,26]. Furthermore, unanchored K48-linked polyubiquitin chains stimulate the IKKi-dependent antiviral responses [27]. However, it is unknown whether IRF3 is a substrate of unanchored polyubiquitin and whether unanchored ubiquitination modifications regulate IRF3-dependent IFN-I production and thus antiviral innate immunity.

USP5 (also known as isopeptidase T) belongs to the ubiquitin-specific protease (USP) family of DUBs, not only removes anchored ubiquitin (Ub) from the target proteins but also uniquely removes and recycles unanchored Ub from the substrates [28,29]. Although USP5 has been reported to function as a scaffold in regulating innate immunity [30], further investigation is required to determine whether it has other targets in this context and whether its DUB enzyme activity is involved in modulating host antiviral innate immunity.

Here, we find that USP5 is significantly downregulated in the host cells infected with RNA viruses including influenza A virus (IAV), vesicular stomatitis virus (VSV), and Sendai virus (SeV). USP5 inhibits antiviral immunity in a DUB enzyme activity-dependent manner by targeting IRF3. Mechanistically, USP5 constitutively interacts with IRF3, deconjugating both anchored K63-linked polyubiquitin chains and unanchored K48-linked polyubiquitin chains on IRF3. The dual cleavage activity of USP5 inhibits IRF3 phosphorylation, nuclear translocation, and subsequent IRF3-driven IFN-β transcription during RNA virus infection. These findings describe a critical role of USP5 in maintaining innate immune homeostasis and provide a novel target for the treatment of RNA virus infectious diseases.

## Results

### Downregulation of USP5 during host antiviral immunity in an IFNAR-dependent manner

To investigate the involvement of DUBs in modulating antiviral immune responses, we reanalyzed microarray data (GSE49840) from RNA virus-infected Calu-3 cells and observed a significant downregulation of USP5 expression (Fig 1A). To further validate this finding, we infected A549 cells with various RNA viruses, including VSV, SeV, and influenza virus A/WSN/33 (WSN). Consistently, USP5 expression was markedly suppressed upon viral infection (Fig 1B and 1C). Similar results were observed in mouse peritoneal macrophages (PMs) and bone marrow-derived macrophages (BMDMs) following infections with SeV, VSV, and WSN (Figs 1D and S1A). Interestingly, the transfection of viral mimics, including poly(I:C) and poly(dA:dT), as well as the stimulation with interferon-β (IFN-β), also resulted in a significant downregulation of USP5 expression (Figs 1D and S1A), suggesting that USP5 was downregulated by the IFN-I signaling. To determine the role of IFN-I in regulating USP5 expression, we derived PMs and BMDMs from wild-type (WT) and IFN-I receptor knockout ($Ifnar1^{-/-}$) mice, followed by VSV infection or poly(I:C) transfection. As we expected, the mRNA level of USP5 was markedly decreased over time after VSV infection or poly(I:C) transfection, while the suppression of USP5 expression was completely abrogated in the absence of IFNAR1 (Figs 1E, 1F, S1B and S1C). In conclusion, our findings demonstrate that USP5 is downregulated during host immunity against viral infection in an IFNAR-dependent manner.

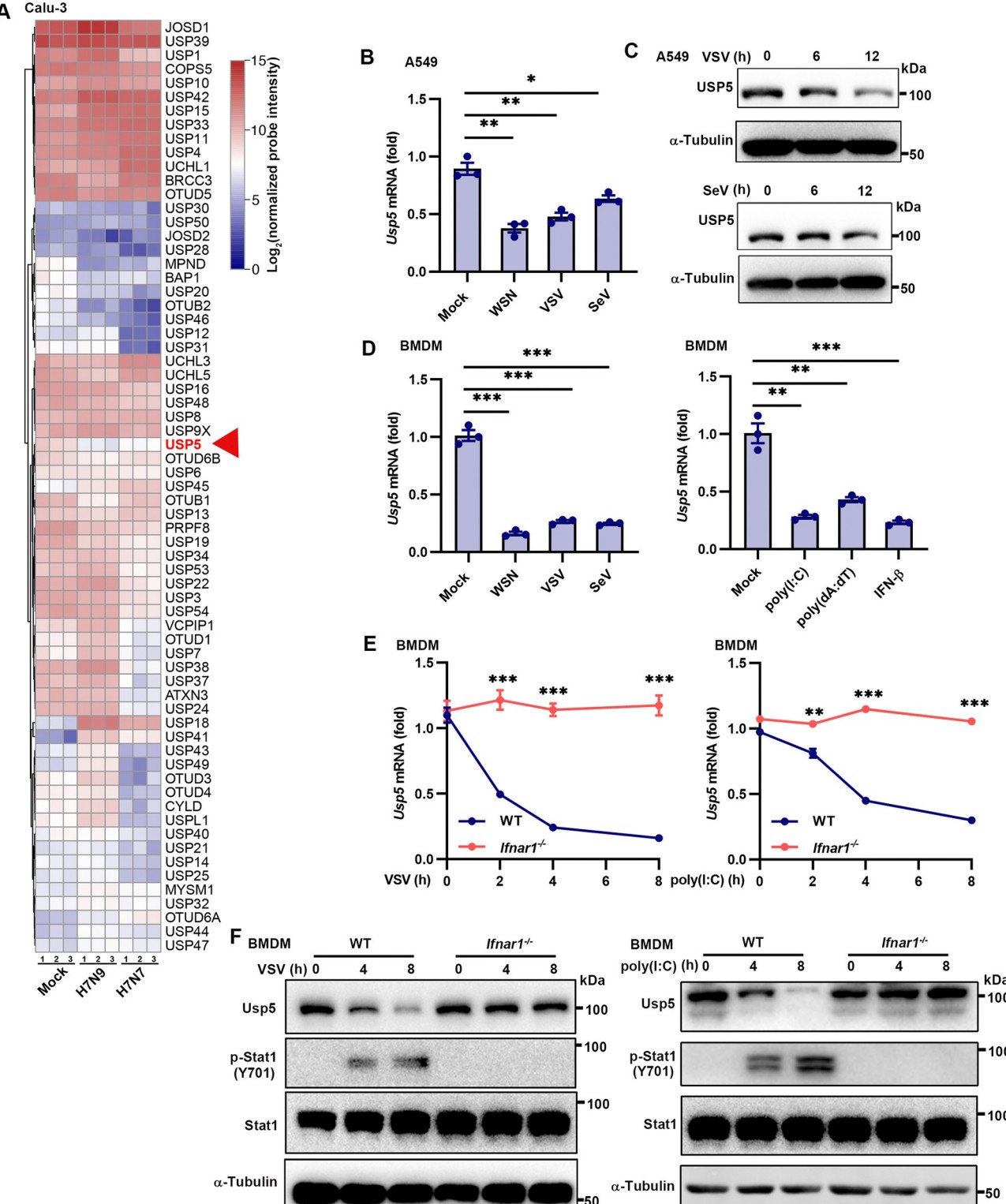

**Fig 1. Downregulation of USP5 during host antiviral immunity in an IFNAR-dependent manner. (A)** The DUB mRNA expression of the IAV-infected or uninfected Calu-3 cells. Relative mRNA expression levels were calculated from the microarray datasets (GSE49840). **(B)** RT-qPCR analysis of *Usp5* expression in the A549 cells infected with WSN (1 MOI), VSV (0.1 MOI) or SeV (0.1 MOI) for 12 h. **(C)** Immunoblot analysis of USP5 expression in the A549 cells infected with SeV (0.1 MOI) or VSV (0.1 MOI) for the indicated times. **(D)** RT-qPCR analysis of *Usp5* expression in the BMDMs infected with WSN (1 MOI), VSV (0.1 MOI), or SeV (0.1 MOI) for 12 h, transfection with poly(I:C) (1 μg/mL), poly(dA:dT) (1 μg/mL), or stimulation

with IFN-β (500 U/mL) for 6 h. (**E**) RT-qPCR analysis of *Usp5* expression in the BMDMs from WT and *Ifnar1*$^{-/-}$ mice, following VSV infection at 0.1 MOI or poly(I:C) transfection at 1 μg/mL for 0, 2, 4, and 8 h. (**F**) Immunoblot measurement of USP5, p-STAT1, and STAT1 levels in the BMDMs from WT and *Ifnar1*$^{-/-}$ mice, following VSV infection at 0.1 MOI or poly(I:C) transfection at 1 μg/mL for 0, 4, and 8 h. Data are representative of 3 independent experiments (B-F). Mean ± SEM, statistical analysis was performed using unpaired two-tailed Student's *t*-test (E) or one-way ANOVA (B and D), \*$p<0.05$, \*\*$p<0.01$, and \*\*\*$p<0.001$ indicate the significant differences.

## USP5 promotes RNA virus infection

To assess the role of USP5 in regulating antiviral innate immunity, we overexpressed USP5 in cell lines and challenged these cells with SeV-GFP or VSV-GFP. Much more green fluorescent protein (GFP)-positive cells and more GFP fluorescence were observed upon USP5 overexpression (Figs 2A and S2A and S2B). Meanwhile, we utilized CRISPR-Cas9 system to efficiently knockout the USP5 gene in A549 cells (S2C and S2D Fig). Following infection with SeV-GFP and VSV-GFP, we observed fewer virus-infected cells in the *USP5*$^{-/-}$ cells compared to the WT cells (Figs 2B and S2E and S2F). In addition, we stably overexpressed USP5 in A549 cells using a lentiviral gene delivery system (S2G Fig). More viral genes and supernatant viral particles of SeV, VSV, and WSN were detected from the USP5 overexpressed cells than the control cells (Fig 2C and 2D). Consistently, we observed fewer viral genes and supernatant viral particles of these RNA viruses from the *USP5*$^{-/-}$ cells compared to the WT cells (Fig 2E and 2F). Taken together, these results have shown that USP5 facilitates RNA virus infection.

## USP5 inhibits host anti-RNA viral innate immunity

We sought to explore the underlying molecular mechanism for the protective role of USP5 deficiency during RNA virus infection. First, we analyzed the production of *IFNB*, *ISG54*, and *CCL5* in USP5-overexpressed and *USP5*$^{-/-}$ A549 cells after viral infection. Less induction of the *IFNB*, *ISG54*, and *CCL5* transcripts was observed in the SeV-infected USP5 overexpressed A549 cells than the control cells (Fig 3A). Conversely, USP5 deficiency facilitated the virus infection-triggered transcription of *IFNB*, *ISG54*, and *CCL5* (Fig 3B). To assess the function of USP5 deficiency in mouse cell lines, we designed a small interfering RNA targeting *Usp5* in RAW264.7 cells to efficiently knock down Usp5 expression (S3A Fig). We observed that Usp5 knockdown markedly promoted SeV- and VSV-induced IFN-β production (S3B and S3C Fig). Next, we investigated the impact of USP5 on the activation of IFN-I signaling pathways during RNA virus infection. We found that USP5 overexpression inhibited phosphorylation of IRF3 (upstream of IFN-β) and STAT1 (downstream of IFN-β) in response to SeV infection (Fig 3C). In contrast, loss of USP5 promoted phosphorylation of IRF3 and STAT1 under the same conditions (Fig 3D). In addition, upon SeV infection, we observed decreased IRF3 nuclear localization following USP5 overexpression, while knockout of USP5 facilitated IRF3 nuclear localization (Fig 3E–3G). Together, these results demonstrate that USP5 negatively regulates innate immune responses against RNA virus infection.

## USP5 interacts with IRF3 via the DNA-binding domain of IRF3

Recent studies are highlighting the importance of the cyclic GMP-AMP synthase (cGAS)-stimulator of interferon genes (STING) signaling axis during RNA virus infection and disease pathogenesis [31,32]. To determine whether USP5 regulates anti-RNA viral innate immunity via the cGAS-STING pathway, we generated USP5 and STING double knockout (*Usp5*$^{-/-}$*Sting*$^{-/-}$) iBMM cells using the CRISPR-Cas9 system (S4A Fig). The results showed that STING knockout did not significantly affect the antiviral ability of USP5 deficiency (S4A and S4B Fig), suggesting that USP5 impairs anti-RNA virus immune responses mainly through the RIG-I signaling pathway rather than the cGAS-STING axis. To explore the role of USP5 in regulating RIG-I signaling, we assessed the roles of USP5 on RIG-I-, MAVS-, TBK1-, IKKi-, and

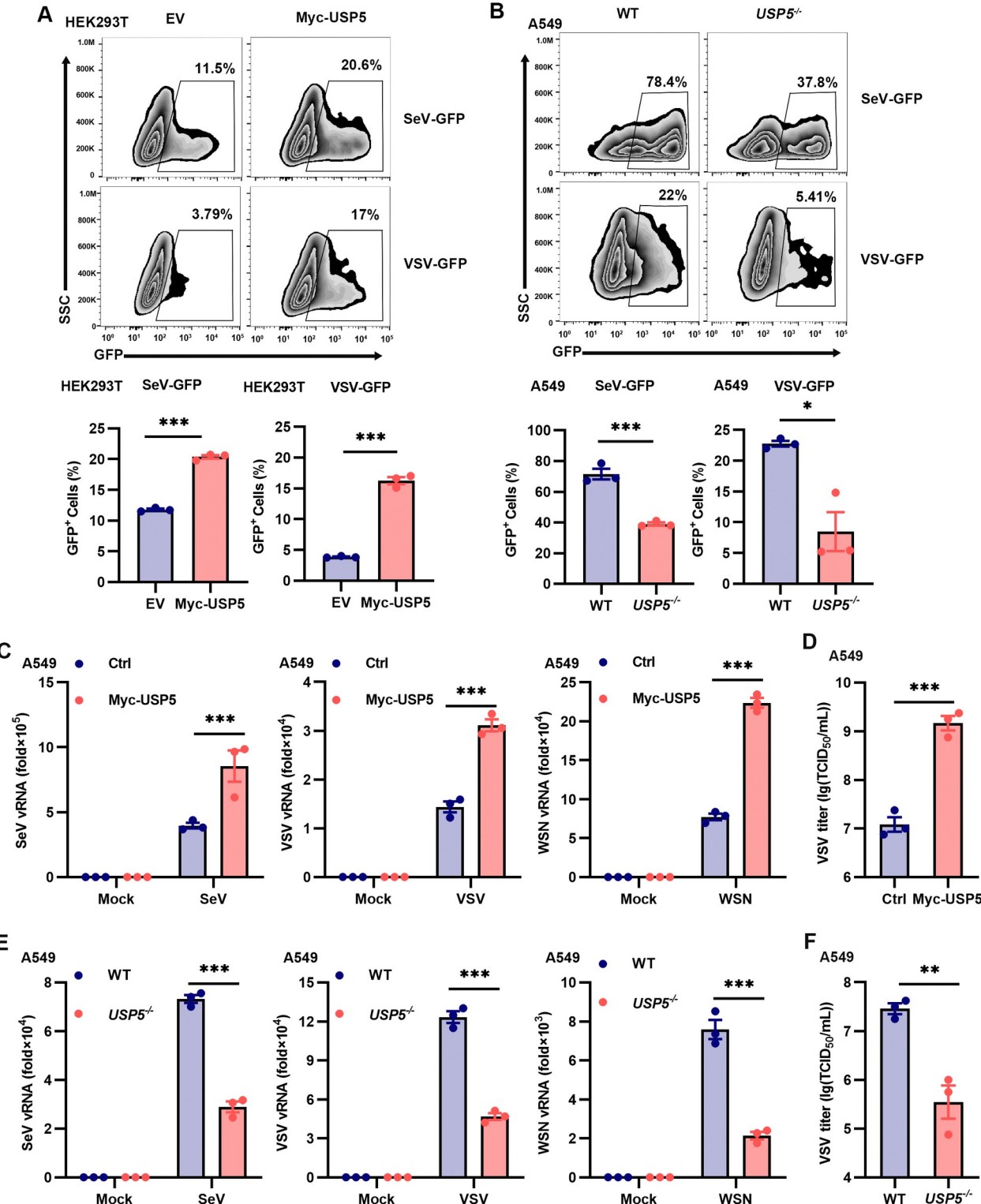

**Fig 2. USP5 promotes RNA virus infection. (A)** Flow cytometry analysis of HEK293T cells transfected with empty vector (EV) or Myc-USP5 for 24 h and infected with SeV-GFP (0.1 MOI) or VSV-GFP (0.1 MOI) for another 6 h. Results from 3 independent experiments were quantified using histograms. **(B)** Flow cytometry analysis of the percentage of GFP-positive cells in the WT and *USP5*$^{-/-}$ A549 cells infected with SeV-GFP (0.5 MOI) or VSV-GFP (0.5 MOI) for 6 h. Results from 3 independent experiments were quantified using histograms. **(C)** RT-qPCR analysis of virus replication in the Ctrl and Myc-USP5 overexpressed A549 cells, following infection with SeV (0.1 MOI), VSV (0.1 MOI), or WSN (1 MOI) for 24 h. **(D)** TCID$_{50}$ assay analysis of viral load in the Ctrl and Myc-USP5 overexpressed A549 cells, following infection with VSV (0.1 MOI) for 24 h. **(E)**

RT-qPCR analysis of virus replication in WT and $USP5^{-/-}$ A549 cells, following infection with SeV (0.1 MOI), VSV (0.1 MOI), or WSN (1 MOI) for 24 h. **(F)** $TCID_{50}$ assay analysis of viral load in the WT and $USP5^{-/-}$ A549 cells, following infection with VSV (0.1 MOI) for 24 h Data are representative of 3 independent experiments (A-F). Mean ± SEM, statistical analysis was performed using unpaired two-tailed Student's $t$-test (A-F), $*p<0.05$, $**p<0.01$, and $***p<0.001$ indicate the significant differences.

IRF3-driven IFN-β transcription and ISGs induction using IFN-β luciferase (IFN-β-luc) and IFN-stimulated response element luciferase (ISRE-luc) reporters. Our findings demonstrated that USP5 significantly inhibited RIG-I-, MAVS-, TBK1-, IKKi-, and IRF3-induced IFN-β and ISRE transcription activities (Fig 4A), suggesting that IRF3 might be the target by which USP5 negatively regulates anti-RNA viral response. We used the bimolecular luminescence complementation (BiLC) assay to analyze whether there was an interaction between USP5 and IRF3 (Fig 4B). We found that USP5 was associated with IRF3, and poly(I:C) transfection further promoted this interaction (Fig 4C). Next, we employed the *in situ* proximity ligation assay (PLA), which visualizes endogenous protein-protein interactions and the subcellular localization of interacting proteins *in situ* [33,34], to investigate the endogenous interaction between USP5 and IRF3 (S4C Fig). We found that USP5 interacted with IRF3, and SeV infection significantly enhanced the endogenous interaction between USP5 and IRF3 in A549 cells (Fig 4D and 4E). To identify the IRF3 domain responsible for binding USP5, we generated a series of IRF3 domain deletion plasmids (Figs 4F and S4D). When the DNA-binding domain of IRF3 was deleted, IRF3 completely lost its ability to bind to USP5 (Fig 4G and 4H). These findings suggest that USP5 interacts with IRF3, and the DNA-binding domain of IRF3 mediates this interaction.

## USP5 impairs both K48-linked and K63-linked polyubiquitination of IRF3

To investigate the involvement of USP5's deubiquitinase activity in regulating antiviral immunity, we constructed plasmids encoding a C335A mutant (USP5-C335A) and a truncated mutant lacking two Ubiquitin-Associated (UBA) domains (USP5-ΔUBA) (Fig 5A). The C335 residue is crucial for the enzymatic activity of USP5, while the UBA domains are primarily responsible for recognizing polyubiquitin chains [29,35]. The USP5-C335A and USP5-ΔUBA mutations did not affect the binding between USP5 and IRF3 but abolished USP5-mediated inhibition of IRF3-induced IFN-β-luc transcriptional activity (Fig 5B–5D). Additionally, these mutations restricted USP5's ability to enhance VSV-Rluc virus replication (Fig 5E). These results indicate that USP5 regulates host antiviral immunity via its DUB activity.

Next, we investigated the effect of USP5 on the total ubiquitination of IRF3 and found that USP5 could significantly remove polyubiquitin chains from IRF3 (Fig 5F). There are seven types of polyubiquitination linkages involving protein degradation, activation, or sub-cellular localization [36]. Further experiments showed that USP5 mainly removed K48- and K63-linked polyubiquitination of IRF3, suggesting that USP5-mediated regulation of two linkages in IRF3 ubiquitination might control different aspects of IRF3 function (Figs 5G and S4E). Moreover, mutations in USP5 (C335A and ΔUBA) abolished its ability to deubiquitinate IRF3 (Fig 5H). The K63-linked polyubiquitination of IRF3 is crucial for its activation and establishment of antiviral capability [20]. Thus, during RNA virus infection, USP5 effectively removes K63-linked polyubiquitination from IRF3 via its DUB enzyme activity, thereby limiting IRF3 transcriptional activities and suppressing antiviral responses.

## USP5 deconjugates K48-linked unanchored and K63-linked anchored polyubiquitin on IRF3

K48-linked polyubiquitination is typically associated with protein degradation via the proteasome, and it is well-established that this process promotes IRF3 degradation [37]. However,

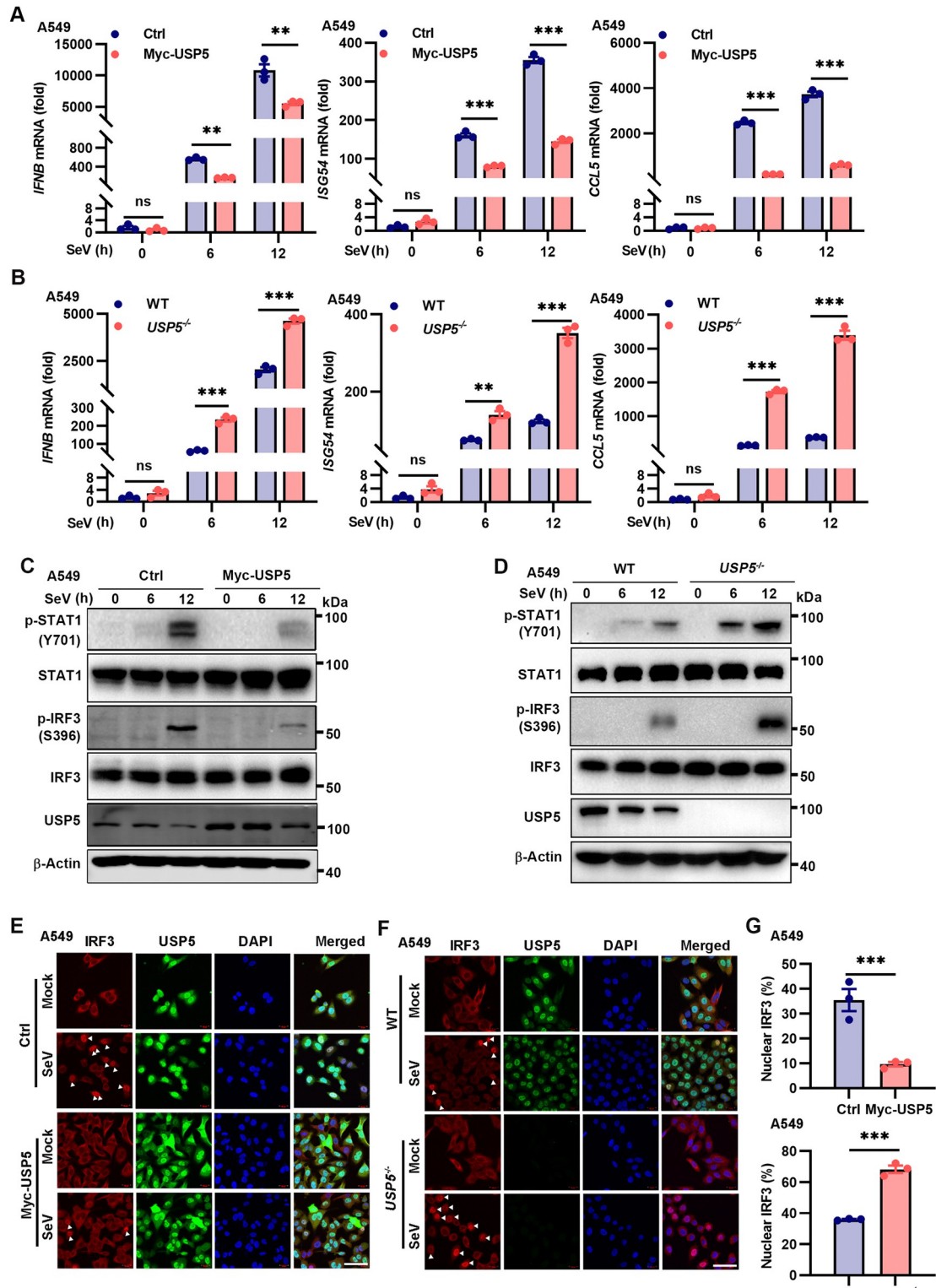

**Fig 3. USP5 inhibits host anti-RNA viral innate immunity. (A)** RT-qPCR analysis of *IFNB*, *ISG54*, and *CCL5* in the Ctrl and Myc-USP5 overexpressed A549 cells, following infected with SeV (0.1 MOI) for 0, 6, and 12 h. **(B)** RT-qPCR analysis of *IFNB*, *ISG54*, and *CCL5* in the WT and *USP5⁻/⁻* A549 cells, following infected with SeV (0.1 MOI) for 0, 6, and 12 h. **(C)** Immunoblot analysis of indicated proteins in the lysates of Ctrl and Myc-USP5 overexpressed A549 cells infected with SeV (0.1 MOI) for 0, 6, and 12 h. **(D)** Immunoblot analysis of indicated proteins in the WT and *USP5⁻/⁻* A549 cells infected with SeV (0.1 MOI) for 0, 6,

and 12 h. **(E)** Immunofluorescence (IF) analysis of the translocation of IRF3 in the Ctrl and Myc-USP5 overexpressed A549 cells infected with or without SeV (0.1 MOI) for 6 h. Scale bar, 50 μm. **(F)** IF analysis of the translocation of IRF3 in the WT and *USP5*$^{-/-}$ A549 cells infected with or without SeV (0.1 MOI) for 6 h. Scale bar, 50 μm. **(G)** The results from 3 independent experiments shown in E and F were quantified using histograms. Data are representative of 3 independent experiments (A-G). Mean ± SEM, statistical analysis was performed using unpaired two-tailed Student's *t*-test (A, B, and G), ns, not significant; **$p<0.01$ and ***$p<0.001$ indicate the significant differences.

K48-linked unanchored polyubiquitin chains enhance IKKi activation during host innate immunity against RNA viruses [27]. Given that USP5 is a specific DUB capable of removing unanchored polyubiquitin chains [29], we investigated its effect on the unanchored ubiquitination of IRF3. Our results showed that USP5 selectively removed K48-linked unanchored polyubiquitin chains from IRF3 without affecting K48-linked anchored polyubiquitin chains (Fig 6A). Conversely, USP5 did not affect K63-linked unanchored polyubiquitin chains on IRF3 but significantly removed K63-linked anchored polyubiquitin chains from IRF3 (Fig 6B). To investigate the regulatory role of K48-linked unanchored ubiquitination in IRF3-mediated antiviral immunity, we constructed an Ub(K48)-G76A plasmid. This site-specific mutant contains a glycine-to-alanine substitution at the C-terminus (G76) of Ub(K48), promoting the generation of unanchored monoubiquitin or polyubiquitin chains due to its reduced ability to form a stable ternary complex catalyzed by E1 [38–40]. Subsequently, we assessed the impact of K48-linked unanchored ubiquitin modification on IRF3-mediated IFN-β transcription and ISG induction using IFN-β-luc and ISRE-luc reporters. We found that Ub(K48)-G76A significantly enhanced IRF3-induced IFN-β and ISRE transcription activities (Fig 6C).

In summary, our findings demonstrate that upon RNA virus infection, USP5 binds to IRF3 and efficiently removes both K63-linked anchored and K48-linked unanchored polyubiquitin chains from IRF3. This dual deconjugating action inhibits IRF3 activation and the subsequent establishment of antiviral states (Fig 7). Therefore, IFN-I-mediated USP5 suppression enhances the K63-linked anchored and K48-linked unanchored ubiquitination of IRF3, thereby promoting IRF3 activation and further activating IFN-I and antiviral genes transcription.

## Discussion

Preformed unanchored polyubiquitin chains exist in the cytoplasm, and certain topologies of some unanchored polyubiquitin chains may function as second messengers, rapidly assembling in response to various stimuli [27,41,42]. This property is crucial for host requirements to rapidly establish an antiviral state to combat viral infections. The role of unanchored polyubiquitin in defending against RNA virus infections has been well-established. K63-linked unanchored polyubiquitination of RIG-I and MDA5 is essential for the assembly of ordered complexes and IFN-I signaling activation [26,43,44]. Consistently, K48-linked unanchored polyubiquitination of IKKi enhances RIG-I signaling and induces ISG production during RNA virus infection [27]. In this study, our findings have shown the pivotal role of K48-linked unanchored polyubiquitination of IRF3 in regulating IRF3-mediated IFN-β production and establishment of antiviral states. Our findings not only identify IRF3 as a novel substrate for unanchored polyubiquitination but also provide evidence that the function of K48-linked polyubiquitination is influenced by the type of linkage (covalent or non-covalent).

The diversity of polyubiquitin chains is a major determinant regulating various biological functions, with the type of polyubiquitin chain linked determining the specific outcomes of the modification [45]. Previous studies have demonstrated that ubiquitin generates seven different types of polyubiquitin chains (K6, K11, K27, K29, K33, K48, and K63), among which the functions of K48 and K63 linkage polyubiquitin chains have been well characterized

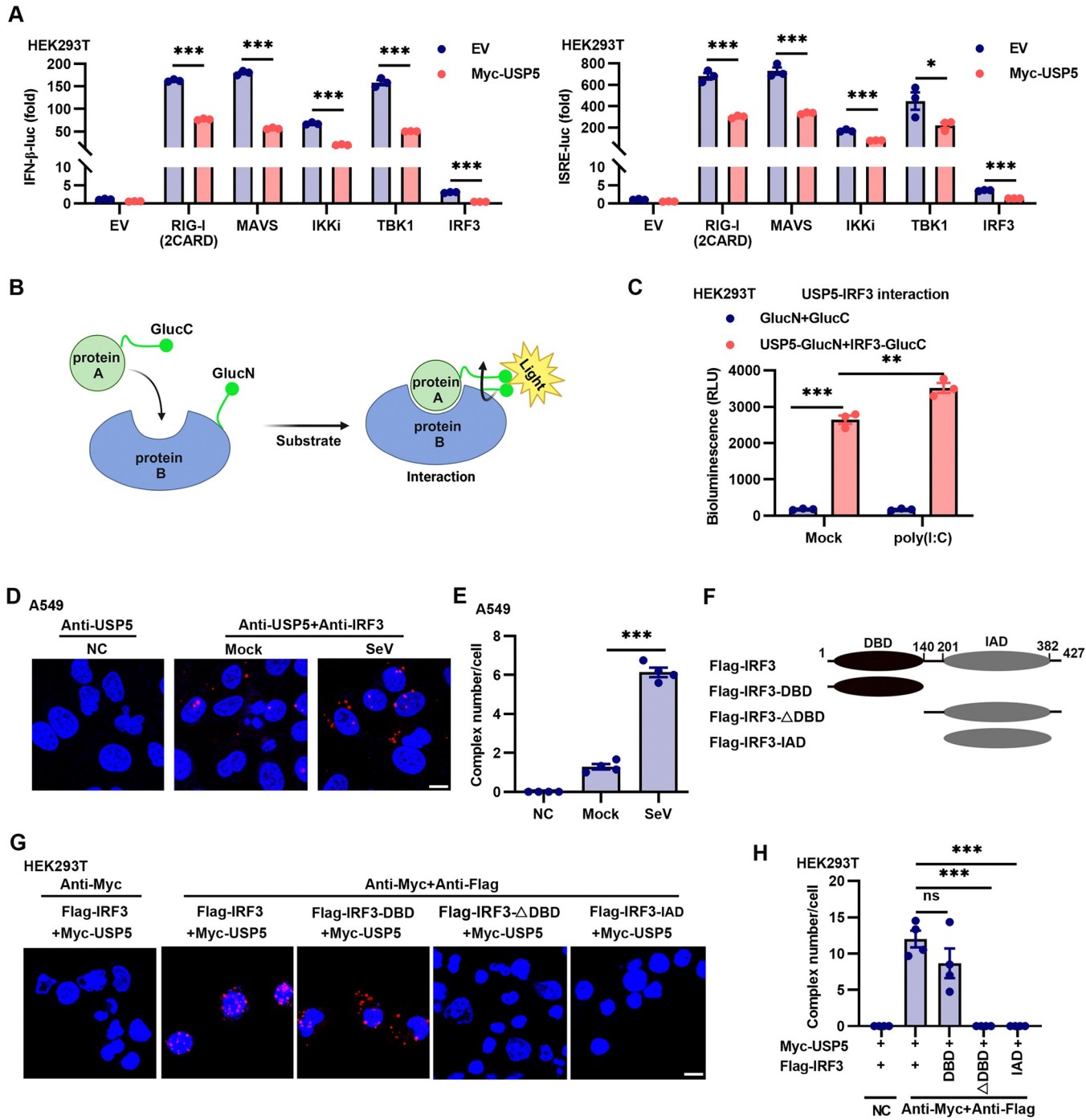

**Fig 4. USP5 interacts with IRF3. (A)** The dual-luciferase activity assay of the IFN-β or ISRE reporter activity in the HEK293T cells co-transfected with either an EV or a USP5-expressing plasmid (200 ng), along with plasmids encoding RIG-I (2CARD), MAVS, IKKi, TBK1, and IRF3 (100 ng each) for 24 h. **(B)** Schematic diagram of the protein complementation assays based on two subdomains of the Gaussia luciferase, GlucC/GlucN. Created with Biorender.com. **(C)** HEK293T cells were transfected with the plasmids GlucC and GlucN or USP5-GlucN and IRF3-GlucC for 24 h and then transfected with or without poly(I:C) for 6 h, luciferase activity was measured from the cell lysates. **(D and E)** *In situ* Proximity Ligation Assay (PLA) analysis of the USP5 and IRF3 interaction in A549 infected with or without SeV (0.1 MOI) for 6 h. The results from 4 representative images were quantified using histograms. Scale bar, 5 μm. **(F)** Schematic diagram of IRF3 domains and truncation mutants. **(G and H)** *In situ* PLA analysis was conducted to map the USP5 interacted domain of IRF3. HEK293T cells were co-transfected with USP5 and full-length IRF3 or various IRF3 truncation mutants (DBD, ΔDBD, IAD) for 24 h, then infected with SeV at an MOI of 0.1 for 6 h, complex (red) represents IRF3 interacted with USP5, DAPI for cell nuclei (blue); The number of complexes per cell was determined by analyzing at least 4 images. Scale bar, 5 μm. Data are representative of 3 independent experiments (A and C). Mean ± SEM, statistical analysis was performed using unpaired two-tailed Student's *t*-test (A and E) or one-way ANOVA (C and H), *$p<0.05$, **$p<0.01$, and ***$p<0.001$ indicate the significant differences.

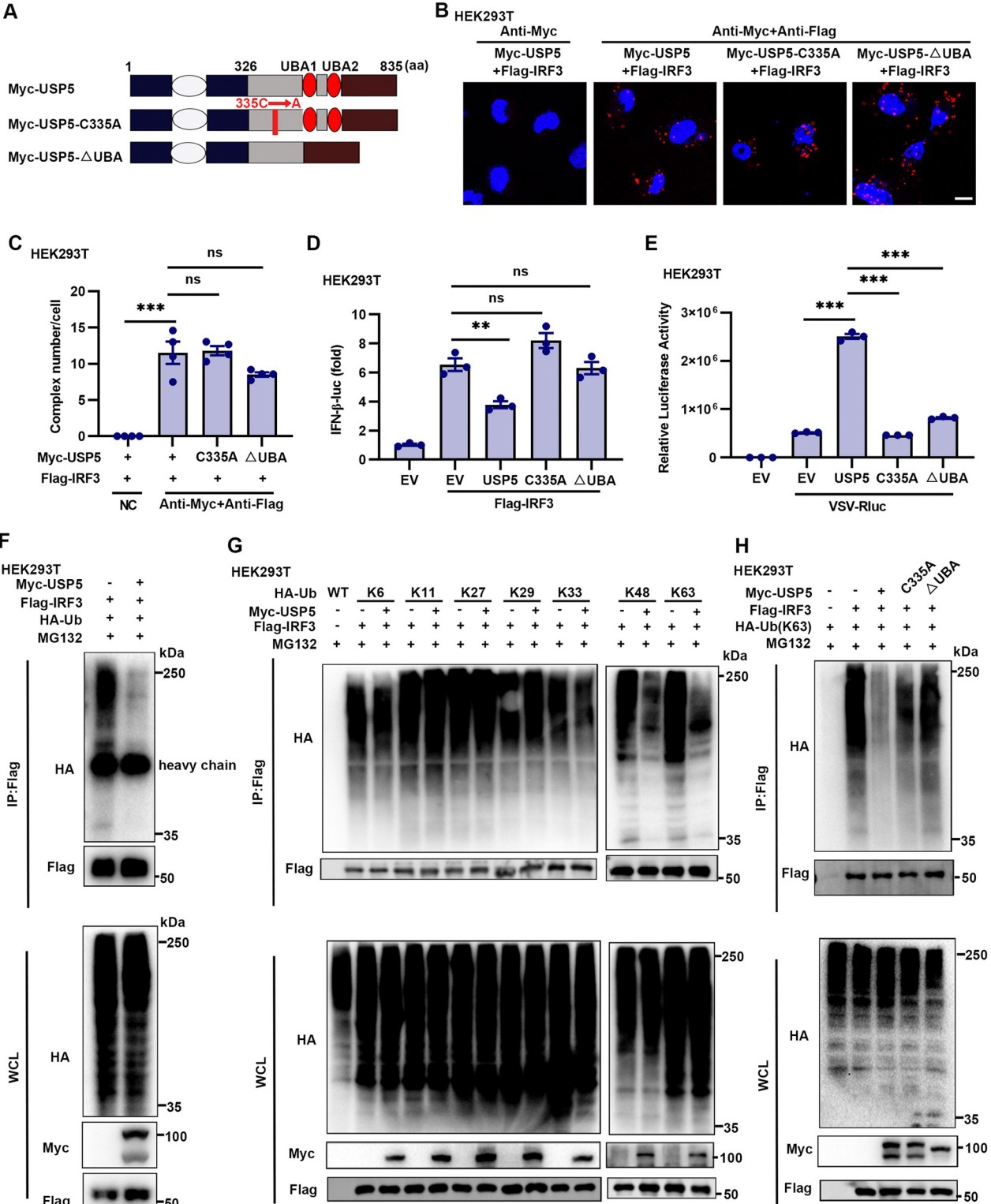

**Fig 5. USP5 deconjugates both K48-linked and K63-linked polyubiquitination of IRF3. (A)** Schematic diagram of USP5 domains and mutants. **(B and C)** *In situ* PLA analysis was conducted to evaluate the interaction between USP5 mutants and IRF3. HEK293T cells were co-transfected with IRF3 and USP5 (full length) or USP5-C335A or USP5-ΔUBA plasmids for 24 h, then infected with SeV (0.1 MOI) for 6 h. The number of complexes per cell was determined by analyzing at least 4 images. Scale bar, 5 μm. **(D)** Dual-luciferase activity of an IFN-β reporter in the HEK293T cells transfected with IRF3 (100 ng) together with USP5, USP5-C335A, or USP5-ΔUBA (200 ng each plasmid). **(E)** HEK293T cells were transfected with

EV, USP5, USP5-C335A, or USP5-ΔUBA plasmids for 24 h, then infected with or without VSV-Rluc for 8 h. Luciferase activity was measured from the cell lysates. **(F)** Immunoblot analysis of the ubiquitination change of IRF3 after USP5 transfection. HEK293T cells were co-transfected with EV, Flag-IRF3, and HA-Ub, or Myc-USP5, Flag-IRF3, and HA-Ub. At 24 h after transfection, cells were treated with MG132 (10 μM) for 8 h. Proteins in the lysates were immunoprecipitated with anti-Flag antibody. **(G)** HEK293T cells were co-transfected with EV, Flag-IRF3, and HA-Ub (K6, K11, K27, K29, K33, K48, K63), or Myc-USP5, Flag-IRF3, and HA-Ub (K6, K11, K27, K29, K33, K48, K63). At 24 h after transfection, cells were treated with MG132 for 8 h. Proteins in the lysates were immunoprecipitated with anti-Flag antibody. **(H)** Immunoblot analysis of the K63-linked ubiquitination change of IRF3 after USP5 transfection. HEK293T cells were co-transfected with EV/Myc-USP5/Myc-USP5-C335A/Myc-USP5-ΔUBA, Flag-IRF3, and HA-Ub(K63). At 24 h after transfection, the cells were treated with SeV and MG132 for 8 h. Proteins in the lysates were immunoprecipitated with anti-Flag antibody. Data are representative of 3 independent experiments (D-H). Mean ± SEM, statistical analysis was performed using one-way ANOVA (C-E), $^{**}p<0.01$ and $^{***}p<0.001$ indicate the significant differences.

[45,46]. K63 polyubiquitin chains are necessary to activate the IFN-I innate immune response. Upon RNA virus infection, E3 ligase TRIM25 enhances the K63 polyubiquitin chains to catalyze the activation of RIG-I [47]. Additionally, K63-linked polyubiquitination of MAVS and TBK1 promotes their activation [48,49]. Evidence has shown that Ubc5-mediated K63-linked polyubiquitination plays a key role in IRF3 activation [20]. These findings underscore the

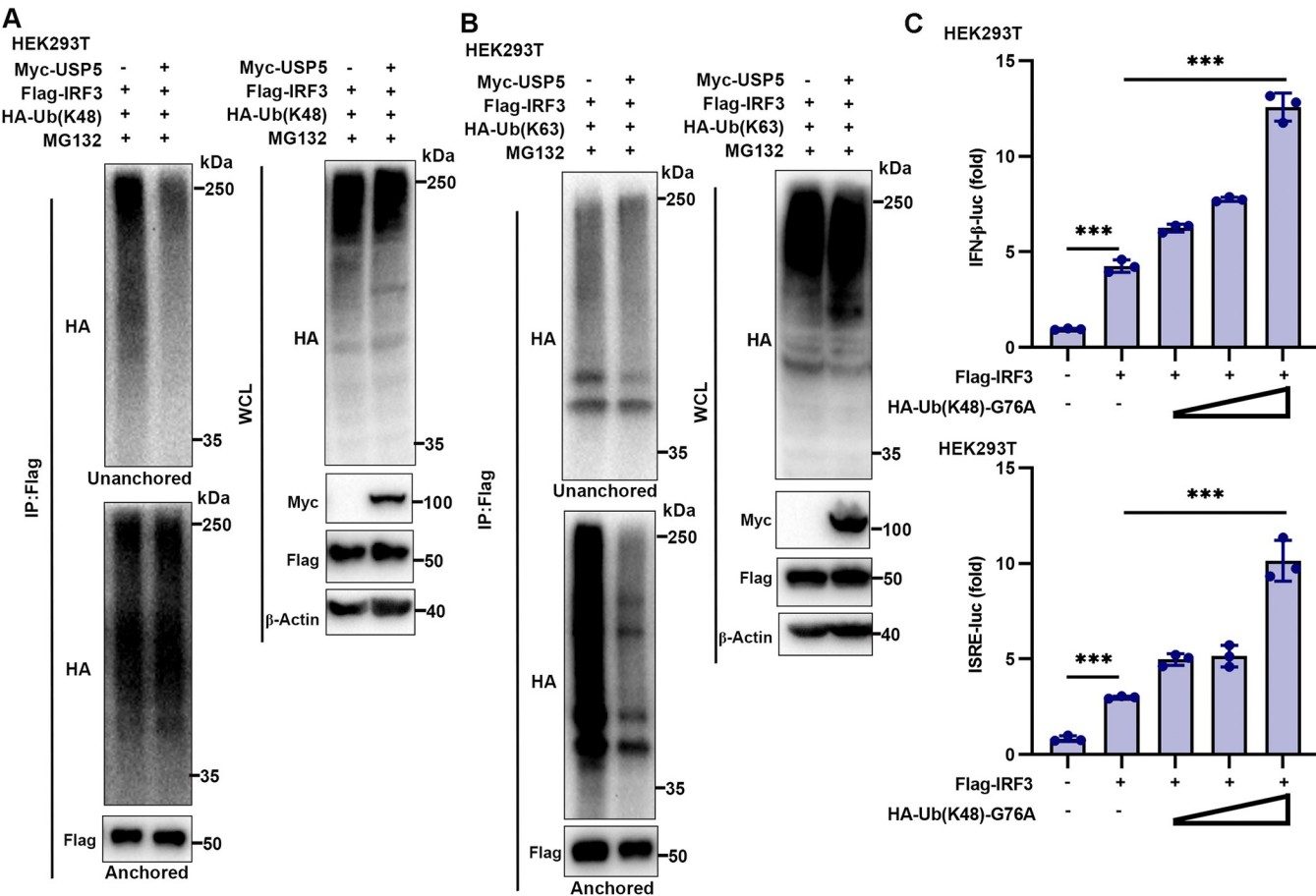

**Fig 6. USP5 deconjugates K48-linked unanchored and K63-linked anchored polyubiquitin on IRF3. (A)** Immunoblot analysis of the K48-linked unanchored ubiquitination change of IRF3 after USP5 transfection. HEK293T cells were co-transfected with EV, Flag-IRF3, and HA-Ub(K48), or Myc-USP5, Flag-IRF3, and HA-Ub(K48). At 24 h after transfection, cells were treated with MG132 for 8 h. Proteins in the lysates were immunoprecipitated with anti-Flag antibody. **(B)** Immunoblot analysis of the K63-linked unanchored ubiquitination change of IRF3 after USP5 transfection. HEK293T cells were co-transfected with EV, Flag-IRF3, and HA-Ub(K63), or Myc-USP5, Flag-IRF3, and HA-Ub(K63). At 24 h after transfection, cells were treated with MG132 for 8 h. Proteins in the lysates were immunoprecipitated with anti-Flag antibody. **(C)** Dual-luciferase activity of the IFN-β and ISRE reporters in HEK293T cells transfected with IRF3 (100 ng) together with different doses (50 ng, 100 ng, and 200 ng) of HA-Ub(K48)-G76A. Data are representative of 3 independent experiments (A-C). Mean ± SD, statistical analysis was performed using one-way ANOVA (C), $^{***}p<0.001$ indicate the significant differences.

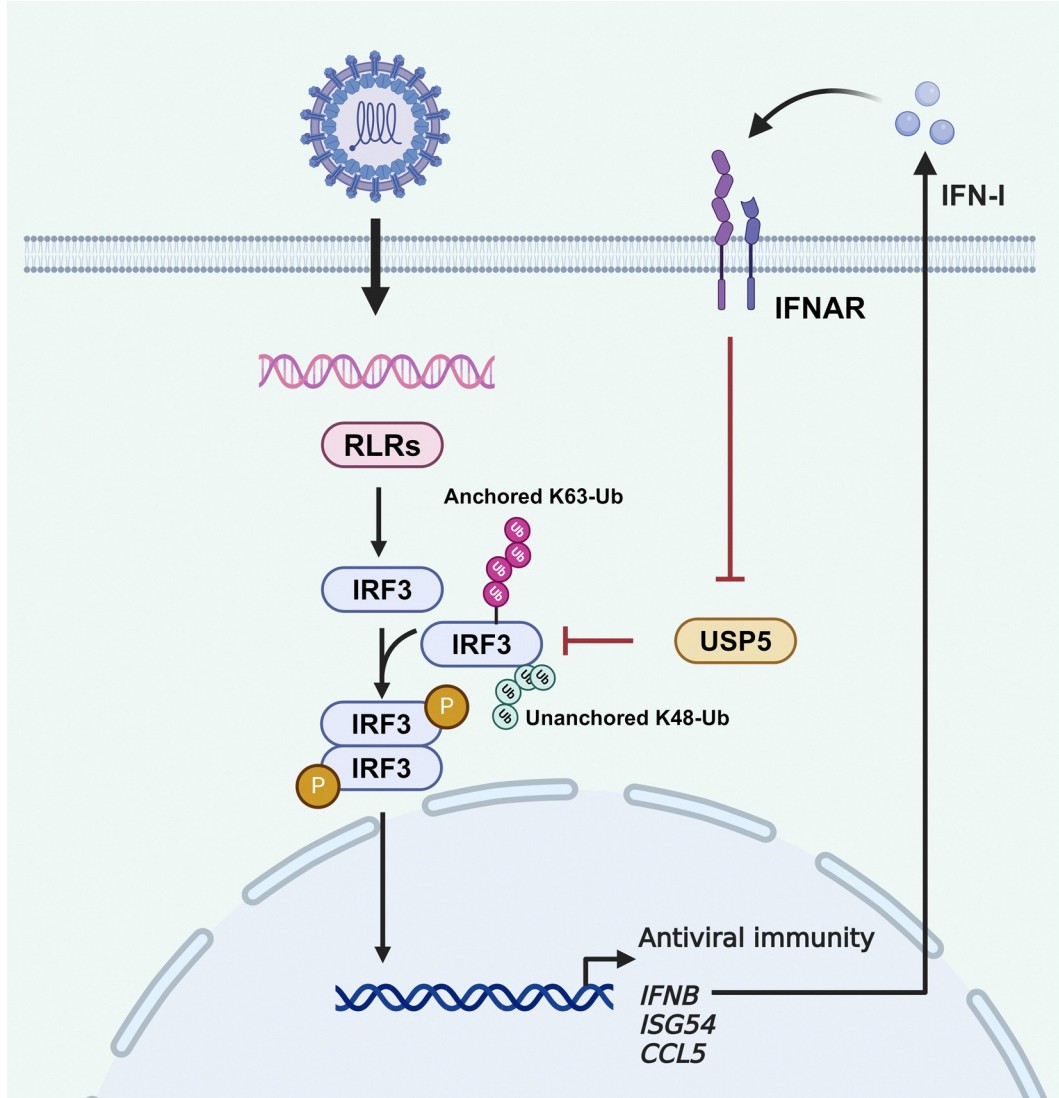

**Fig 7. Working model.** Upon RNA virus infection, a large amount of IFN-I is induced, leading to the downregulation of USP5 expression. USP5 binds to IRF3 and efficiently removes both K63-linked anchored polyubiquitin chains and K48-linked unanchored polyubiquitin chains from IRF3. Therefore, IFN-I-mediated USP5 suppression enhances the K63-linked anchored and K48-linked unanchored ubiquitination of IRF3, thereby promoting IRF3 activation and further activating IFN-I and antiviral genes transcription. Created with Biorender.com.

increasing significance of K63-linked ubiquitination in innate immune signaling. However, our understanding of the regulatory mechanism governing K63-linked ubiquitination of IRF3 remains limited. Although the DUB OTUD1 has been shown to remove K63-linked polyubiquitination in autoimmune disease models, its involvement in early innate immune responses by reducing IRF3 K63 ubiquitination is minimal, primarily due to its upregulation during viral infection [21]. In our study, we observed that IFN-I production during early stage of innate immune responses limits USP5 expression. Additionally, we identified USP5 as a DUB capable of effectively removing the K63-linked ubiquitination of IRF3. We also found that K48-linked unanchored polyubiquitin has a similar function to K63-linked anchored polyubiquitin in triggering IRF3 activation. Future research should ideally include a detailed study to evaluate which form of ubiquitination is more crucial for IRF3 activation. Although we have identified

IRF3 as a target of USP5 in regulating antiviral responses, our study does not exclude the involvement of other USP5 substrates in regulating host antiviral innate immunity.

USP5 is widely expressed in normal tissues and cells, and its roles in suppressing p53 transcriptional activity, promoting DNA damage repair, and regulating tumor cell growth highlight its critical involvement in maintaining normal cellular functions [50–52]. Therefore, a clearer understanding of the mechanisms that govern USP5 expression is required. The transcription factor EBF1 enhances the transcription of USP5 in colorectal cancer cells by binding to its promoter [53]. In keloid fibroblasts, the polypyrimidine tract-binding protein (PTB) controls cell proliferation by regulating the alternative splicing of USP5 [54]. While the regulator of USP5 in antiviral immune responses remains unclear. In our study, we have identified IFN-I and its activators, including poly(I:C) and poly(dA:dT), as the upstream suppressors of USP5, and identified IFNAR1 as a key receptor regulating USP5 expression.

In summary, we have demonstrated that IFN-I-mediated USP5 suppression forms an alternative regulation pathway to facilitate host defense against numerous RNA viruses. Upon RNA virus infection, IFN-I production is induced, leading to the suppression of USP5 expression. This suppression enhances K63-linked anchored and K48-linked unanchored ubiquitination of IRF3, thereby promoting IRF3 activation and further activating IFN-I transcription. However, DNA viruses can also initiate the expression of IFN-I through the cGAS-STING signaling pathway [55]. Our findings have shown that IFN-I significantly suppresses the expression of USP5, indicating that DNA viruses may also suppress USP5 expression. Consequently, during DNA virus infection, USP5 potentially plays a role in regulating the anti-DNA viral immune response. Nevertheless, the function of USP5 in antiviral defense against DNA viruses and the underlying molecular mechanisms involved still require further investigation.

## Materials and methods

### Ethics statement

All animal experiments were conducted according to the US National Institutes of Health Guide for the Care and Use of Laboratory Animals and approved by the Animal Service Center of Suzhou Institute of System Medicine (ISM) (ISM-IACUC-0011-R).

### Mice

The C57BL/6J WT mice (male, 6–8 weeks) were purchased from Beijing Vital River Laboratory Animal Technology Company. The $Ifnar1^{-/-}$ mice with C57BL/6J background were gifted from Genhong Cheng Laboratory (University of California, Los Angeles, CA). All the mice were maintained in the specific pathogen-free (SPF) environment at ISM under controlled temperature (25°C) and a 12 h day-night cycle.

### Reagents

Anti-USP5 antibody (#sc-390943) was purchased from Santa Cruz Biotechnology. Anti-phospho-IRF-3 (Ser396, #29047), anti-IRF3 (#4302 and #11904), anti-phospho-STAT1 (Tyr701, #9167), anti-STAT1 (#14994), anti-HA-Tag-HRP (#2999), anti-Myc (#13987), anti-Flag (#8146), anti-K48-linkage specific polyubiquitin (#12805), anti-K63-linkage specific polyubiquitin (#12930), and anti-β-Actin (#3700) antibodies were from Cell Signaling Technology. Anti-α-Tubulin antibody (#T5168), mouse monoclonal anti-FLAG M2-HRP antibody (#A8592) and rabbit polyclonal anti-c-Myc-HRP antibody (#A5598), and MG132 (#M8699) were from Sigma. Poly(I:C) (LMW) (#tlrl-picw) and poly(dA:dT) (#tlrl-patn-1) were from InvivoGen. The recombinant mouse IFN-β was from PBL Assay Science (Piscataway, NJ).

## Cell culture and activation

HEK293T, A549, and RAW264.7 cells were purchased from the American Type Culture Collection (ATCC). HEK293T and A549 cells were maintained in Dulbecco's modified Eagle's medium (DMEM) (GIBCO) containing 10% heat-inactivated fetal bovine serum (FBS) (GIBCO) and 1% penicillin/streptomycin solution (P/S) (Invitrogen). RAW264.7 cells were maintained in RPMI 1640 medium (GIBCO) containing 10% FBS and 1% P/S solution. Bone marrow-derived macrophages (BMDMs) were isolated from femurs and tibias of mice, and differentiated in the RPMI1640 medium with 10% FBS, 1% P/S, and 1% M-CSF-conditioned medium for 7 days as we described previously [56]. For mouse primary peritoneal macrophage (PM) preparation, the mice (male, 6–8 weeks) were intraperitoneally (*i.p.*) injected with 3% fluid thioglycollate medium (BD). Three days later, peritoneal exudate cells were collected and cultured for 1 h, the medium was replaced, and the adherent monolayer cells were PMs. All cells were cultured at 37˚C in a 10% $CO_2$ atmosphere. To activate the cells, poly(I:C) and poly (dA:dT) were transfected into BMDMs, PMs, or HEK293T cells by Lipofectamine 2000. The ratio of transfection reagent over ligands was 2 (μL/μg). Additionally, IFN-β was directly added into the macrophage culture medium to stimulate the cells.

## Plasmid construction

The Myc-USP5, Myc-USP5-ΔUBA, Myc-USP5-C335A, HA-Ub(K6), HA-Ub(K11), HA-Ub (K27), HA-Ub(K29), HA-Ub(K33), HA-Ub(K48), HA-Ub(K63), and HA-Ub plasmids were kindly provided by Professor Jun Cui (Sun Yat-Sen University, China). The Flag-IRF3 and IRF3 mutants including Flag-IRF3-DBD, Flag-IRF3-ΔDBD, and Flag-IRF3-IAD plasmids were gifted by Dr. Xiaohong Du (ISM). PCR amplification was conducted utilizing pcDNA3.1-Myc-USP5 as the template, followed by insertion into the FG-EH-DEST2-PGK--Puro-WPRE (FGEH) empty vector to generate the FGEH-Myc-USP5 plasmid. The HA-Ub (K48)-G76A plasmid was constructed by the Mut Express II Fast Mutagenesis Kit V2 (Vazyme). All constructs were confirmed by Sanger DNA sequencing and all the amplification primers were available upon request.

## ELISA

The IFN-β protein concentration in the cell supernatant was measured using the ELISA kit specific for IFN-β (PBL Assay Science), following the manufacturer's instructions.

## Luciferase assay

HEK293T cells were plated in 24-well plates and incubated for 24 h at 37˚C and the cells were transfected with a mixture of IFN-β promoter firefly luciferase reporter plasmid (150 ng/well), Renilla luciferase (10 ng/well), together with EV or other indicated plasmids using polyethyleneimine (PEI) transfection reagents. These cells were collected 24 h after transfection and luciferase activity was measured with the Dual-Luciferase Assay (Promega). The relative activities of VSV-Rluc were measured 8 h post-infection using the Renilla luciferase assay system (Promega).

## Bimolecular luminescence complementation (BiLC) assay

BiLC assays were utilized to assess the interaction between USP5 and IRF3. The open reading frames of USP5 and IRF3 were cloned into the BiLC reporter system, generating USP5-GlucN and IRF3-GlucC reporter constructs, as described previously [57]. For the interaction analysis, HEK293T cells were seeded in 24-well plates and transfected with either control vectors or the

USP5-GlucN and IRF3-GlucC constructs. At 24 h post-transfection, the cells were transfected with or without poly(I:C) (1 μg/mL) for 6 h. The relative Gaussia luciferase activities, indicative of USP5-IRF3 interaction, were measured using the Renilla luciferase assay system (Promega).

## Immunoprecipitation

For immunoprecipitation (IP), whole-cell extracts were prepared post transfection. A portion of the total lysate was reserved as an input control. The left lysate was then incubated with anti-Flag or anti-IRF3 agarose beads at 4°C for 2 h. Subsequently, the beads were washed 3 times with lysis buffer and eluted with 1× SDS Loading Buffer in the 100°C metal bath for 10 min. The beads were removed by centrifugation, and the proteins were loaded, run SDS-PAGE gel, and transferred to a PVDF membrane (Millipore). The membrane was further incubated with the indicated antibodies. Protein detection was performed using chemiluminescence (ECL, Millipore) or the LI-COR Odyssey CLx imaging system.

## Ubiquitination analysis

For the detection of protein ubiquitination, we transfected HEK293T cells with the indicated plasmids for 24 h and then treated these cells with MG132 (10 μM) for an additional 8 h. Concurrently, for endogenous ubiquitination assessment, A549 cells were infected with SeV for 1 h, and treated with MG132 (10 μM) for another 7 h. After these procedures, the cells were collected, and whole-cell extracts were prepared. The ubiquitination levels of exogenous or endogenous IRF3 were assessed using the standard IP protocol.

## Unanchored ubiquitination analysis

For the analysis of unanchored polyubiquitination of IRF3, we followed the established protocol for isolating endogenous unanchored polyubiquitin chains on RIG-I and IKKi, as previously described [25,27]. The whole-cell extracts were prepared following the transfection with the indicated plasmids for 24 h. Afterwards, the extracts were incubated with anti-flag agarose beads for 2 h at 4°C. Subsequently, the beads were washed extensively with the lysis buffer at least 3 times. The proteins were immunoprecipitated by incubating the beads at 75°C for 5 min, and the eluted unanchored constituents were analyzed by immunoblot. Concurrently, the anchored immunoprecipitates remaining on the beads were washed 3 times, eluted, and also analyzed by immunoblot. Aliquots of whole-cell lysate (WCL) served as input controls.

## Immunofluorescence assay

A549 cells were seeded in glass-bottom dishes. After infection with SeV for the indicated times, the dishes were collected. The cells were then briefly washed with PBS, fixed with 4% paraformaldehyde, permeabilized with 0.2% Triton X-100, and blocked with 1% goat serum. For the colocalization assay, the cells were incubated with the indicated primary antibodies (1:50 dilution) and subsequently stained with a fluorescent secondary antibody (1:200 dilution). Following incubation with 4',6-diamino-2-phenylindole (DAPI), fluorescence imaging was performed using a LEICA TCA SP8 confocal microscope.

## *In Situ* Proximity ligation assay

Duolink *in situ* PLA (Duolink Detection kit) was used to detect interactions between USP5 and IRF3 (WT or mutants). Cells were fixed using 4% formaldehyde, permeabilized with 0.2% Triton X-100, and incubated with indicated primary antibodies (mouse anti-Usp5 and rabbit anti-IRF3; mouse anti-Flag and rabbit anti-Myc) for assessing USP5-IRF3 interaction [58].

Slides were comprehensively assessed using a LEICA TCS SP8 confocal microscope. The interaction signals within each cell were quantified using ImageJ software.

## Quantitative real-time PCR (RT-qPCR)

Total cellular RNA was extracted using TRIzol reagent (ThermoFisher Scientific), and 500 ng of total RNA was reversely transcribed into cDNA using the PrimeScript RT Master Mix (Takara). TB Green Premix *Ex Taq* (Tli RNaseH Plus) (Takara) was used for RT-qPCR amplification on a Roche LightCycler 480 II system. The RT-qPCR primer sequences for target genes were listed in S1 Table.

## RNA interference in RAW264.7

siRNA oligonucleotides targeting mouse *Usp5* (ID: 22225) (5'-GCUGUGGAAG CCCUAC UUUTT-3') and negative control siRNA (5'-UUCUCCGAACGUGUC ACGUTT-3') were ordered from GenePharma. RAW264.7 cells were transfected with indicated siRNA using INTERFERin (Polyplus-transfection) according to the manufacturer's instructions. The RNAi efficiency was checked by RT-qPCR 36 h after transfection.

## Flow cytometry analysis

HEK293T and A549 cells were collected following infection with the SeV-GFP or VSV-GFP at the indicated MOI and time points. GFP-positive cells were analyzed using a Life Launch Attune NxT Flow Cytometer (Thermo Fisher Scientific, Waltham, MA). Data analysis was conducted using FlowJo software, version 10.0.

## Microarray data analysis

For microarray analysis of IAV (H7N7 and H7N9)-infected or uninfected Calu-3 cells, raw data (accession no. GSE49840) were downloaded from GEO, and the DUB mRNA expression value was normalized by probe intensity.

## Statistical analysis

The number of experimental repeats is shown in the figure legend. All bar graphs are means with either SD or SEM. Statistical analysis was conducted using unpaired two-tailed Student's *t*-test or one-way ANOVA in GraphPad Prism 9 software. Differences between groups were considered significant when the *p*-value was less than 0.05. $^*p<0.05$, $^{**}p<0.01$, and $^{***}p<0.001$.

## Supporting information

**S1 Fig. Downregulation of USP5 during host antiviral immunity in an IFNAR-dependent manner. (A)** RT-qPCR analysis of *Usp5* mRNA expression in the PMs following infection with WSN (1 MOI), VSV (0.1 MOI), and SeV (0.1 MOI) for 12 h, transfection with poly(I:C) (1 μg/mL), poly(dA:dT) (1 μg/mL), or stimulation with IFN-β (500 U/mL) for 6 h. **(B)** RT-qPCR analysis of *USP5* mRNA levels in the PMs from WT and *Ifnar1*$^{-/-}$ mice, following VSV infection at 0.1 MOI or poly(I:C) transfection at 1 μg/mL for 0, 2, 4, and 8 h. **(C)** Immunoblot analysis of *Usp5* mRNA levels in PMs from WT and *Ifnar1*$^{-/-}$ mice, following VSV infection at 0.1 MOI or poly(I:C) transfection at 1 μg/mL for 0, 4, and 8 h. Data are representative of 3 independent experiments (A-C). Mean ± SEM, statistical analysis was performed using unpaired two-tailed Student's *t*-test (B) or one-way ANOVA (A), $^*p<0.05$, $^{**}p<0.01$, and $^{***}p<0.001$ indicate the significant differences.
(TIF)

**S2 Fig. USP5 facilitates RNA virus infection.** **(A)** Fluorescence microscopy analyses of HEK293T cells transfected with EV or Myc-USP5 for 24 h, following infection with SeV-GFP (0.1 MOI) for 8 h. Scale bar, 200 μm. **(B)** Fluorescence microscopy analyses of HEK293T cells transfected with EV or Myc-USP5 for 24 h, following infection with VSV-GFP (0.1 MOI) for 8 h. Scale bar, 200 μm. **(C)** Schematic diagram of the CRISPR/Cas9 gene editing protocol for USP5 and the Sanger sequencing results. **(D)** Immunoblot analysis of USP5 expression in WT and *USP5*[-/-] A549 cells. **(E)** Fluorescence microscopy analyses of WT and *USP5*[-/-] A549 cells infected with SeV-GFP (0.1 MOI) for 6 h. Scale bar, 200 μm. **(F)** Fluorescence microscopy analyses of WT and *USP5*[-/-] A549 cells infected with VSV-GFP (0.1 MOI) for 6 h. Scale bar, 200 μm. **(G)** Immunoblot analysis of Myc-USP5 expression in Ctrl and Myc-USP5 overexpressed A549 cells. Data are representative of 3 independent experiments (A, B, and D-G). Created with Biorender.com.
(TIF)

**S3 Fig. USP5 inhibits host anti-RNA viral innate immunity.** **(A)** RT-qPCR analysis of the RNAi efficiency targeting USP5 in RAW264.7 cells 36 h post-transfection. **(B)** RT-qPCR and ELISA analyses were conducted to assess *Ifnb* mRNA levels and IFN-β protein expression, respectively, in the control and USP5 knockdown RAW264.7 cells at 0, 12, and 24 h post-infection with SeV (0.1 MOI). **(C)** RT-qPCR and ELISA analyses were conducted to assess *Ifnb* mRNA levels and IFN-β protein expression, respectively, in control and USP5 knockdown RAW264.7 cells at 0, 12, and 24 h post-infection with VSV (0.1 MOI). Data are representative of 3 independent experiments (A-C). Mean ± SEM, statistical analysis was performed using unpaired two-tailed Student's *t*-test (A-C), **$p<0.01$, and ***$p<0.001$ indicate the significant differences.
(TIF)

**S4 Fig. USP5 deconjugates both K48-linked and K63-linked polyubiquitination of IRF3.** **(A)** Immunoblot analysis of USP5 and STING expression in WT and *Usp5*[-/-]*Sting*[-/-] iBMMs. **(B)** RT-qPCR analysis of *Ifnb*, *Isg54*, and *Ccl5* in WT and *Usp5*[-/-]*Sting*[-/-] iBMMs, following infection with SeV (0.1 MOI) for 0, 6, and 12 h. **(C)** The schematic diagram indicates the principle of *in situ* PLA measuring endogenous USP5-IRF3 protein interactions in cells. **(D)** Immunoblot analysis of the expression of IRF3 truncation mutants. **(E)** Immunoblot analysis of the effect of USP5 on endogenous K48 and K63 ubiquitination of IRF3 following infection with SeV (0.1 MOI) for 8 h. Data are representative of 3 independent experiments (A, B, D, and E). Mean ± SEM, statistical analysis was performed using unpaired two-tailed Student's *t*-test (B), *$p<0.05$, **$p<0.01$, and ***$p<0.001$ indicate the significant differences. Created with Biorender.com.
(TIF)

**S1 Table. The primer sequence information.**
(XLSX)

**S1 Data. Datasheet containing raw data used to build the graphs in this manuscript.**
(XLSX)

## Acknowledgments

We appreciate the excellent technical support from the RNA technology platform of ISM. We thank Dr. Xiaohong Du and Professor Jun Cui for providing plasmids. We are grateful to Professor Genhong Cheng for supplying the *Ifnar1*[-/-] mice.

## Author Contributions

**Conceptualization:** Xiulong Xu, Feng Ma.

**Data curation:** Zigang Qiao, Dapei Li, Fan Zhang.

**Formal analysis:** Zigang Qiao, Siying Liu.

**Funding acquisition:** Dapei Li, Fan Zhang, Jingfei Zhu, Feng Ma.

**Investigation:** Zigang Qiao, Dapei Li, Fan Zhang, Jingfei Zhu, Siying Liu, Xue Bai.

**Methodology:** Zigang Qiao, Haiping Yao, Zhengrong Chen, Yongdong Yan.

**Resources:** Xiulong Xu, Feng Ma.

**Supervision:** Feng Ma.

**Validation:** Zigang Qiao, Dapei Li, Fan Zhang.

**Visualization:** Zigang Qiao, Feng Ma.

**Writing – original draft:** Zigang Qiao, Fan Zhang, Feng Ma.

**Writing – review & editing:** Zigang Qiao, Fan Zhang, Feng Ma.

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
