## [Decision Letter · Decision Letter 0]

7 Oct 2024

Dear Professor Ma,

Thank you very much for submitting your manuscript "USP5 restricts anti-RNA viral innate immunity by deconjugating K48-linked unanchored and K63-linked anchored ubiquitin on IRF3" for consideration at PLOS Pathogens. As with all papers reviewed by the journal, your manuscript was reviewed by members of the editorial board and by several independent reviewers. There was clear enthusiasm for the work you presented. In light of the reviews (below this email), we would like to invite the resubmission of a significantly-revised version that takes into account the reviewers' comments.

We cannot make any decision about publication until we have seen the revised manuscript and your response to the reviewers' comments. Your revised manuscript is also likely to be sent to reviewers for further evaluation.

Most of the reviewer requests are addressable. However, you will see that reviewer 3 requests the identification of the lysine residue in IRF3 that is modified with ubiquitin. This is a challenging experiment and not needed to support the claims, and thus **this experiment is not required for resubmission. **

Sincerely,

Andrew Mehle

Academic Editor

PLOS Pathogens

Benhur Lee

Section Editor

PLOS Pathogens

Michael Malim

Editor-in-Chief

PLOS Pathogens

orcid.org/0000-0002-7699-2064

Reviewer's Responses to Questions

**Part I - Summary**

Reviewer #1: Here the authors note from published work that the deubiquitinase USP5 is downregulated during RNA virus infections. They confirm this finding in several RNA virus infections and examine whether this has functional consequences. They find that USP5 overexpression increases infections while its knockout promotes infection. They observe an inverse correlation between USP5 expression and IFNb induction, suggesting that USP5 negatively regulates interferon induction during infection. They narrow in on IRF3 by using IFNb promoter reporter assays with overexpression of RIG-I/MDA5 signaling pathway components. Direct interaction of USP5 and IRF3 that is increased by infection was confirmed with several independent assays. Mechanistically, they show that USP5 decreases K63- and K48-linked polyubiquitination of IRF3 and that inhibition of IFNb induction and increase of viral infection is dependent on USP5 catalytic activity. The data are clear and convincing in support of the conclusion that USP5 deubiquitinates IRF3 to suppress its IFN-induction activity. This finding is significant because mechanisms limiting IFN induction are important for preventing excessive inflammation and tissue damage. Aside from minor suggestions and one major issue with data interpretation/writing as noted in point 4 below, I support the publication of this manuscript.

Reviewer #2: The study by Qiao et al. entitled “USP5 restricts an-RNA viral innate immunity by deconjugating K48-linked unanchored and K63-linked anchored ubiquitin on IRF3” described a novel role of USP5 in regulating IRF3-dependent anti-RNA viral innate immunity. They have not only demonstrated a comprehensive regulatory mechanism that USP5 utilizes its dual-cleavage activity to deubiquitinate IRF3, but also provided a potential target for treating RNA virus infectious diseases. Overall, this is an interesting study with convincing results.

Reviewer #3: In this manuscript, the authors reported that the deubiquitinase USP5 is significantly downregulated during the host's innate immune response against RNA viruses, in an IFNAR-dependent manner. USP5 binds to IRF3 and efficiently removes both K63-linked anchored polyubiquitin chains and K48-linked unanchored polyubiquitin chains from IRF3. This action inhibits IRF3-driven transcription of IFN-β and the induction of IFN-stimulated genes (ISGs). The authors further demonstrated that USP5 negatively regulates IRF3-induced antiviral immune responses through its DUB activity. Overall, this is an interesting study with good novelty, which offers valuable insights into the regulation of USP5-mediated immune responses. However, several issues need to be addressed before consideration of publication.

**Part II – Major Issues: Key Experiments Required for Acceptance**

Reviewer #1: (No Response)

Reviewer #2: 1.The authors should add WB results for confirming the overexpression, knockdown, and knockout of USP5 in A549 cells and iBMMs. These results should be displayed adjacently to the corresponding functional experiments or included them in the supplementary figures.

2.In Figures S2A and S2B, the GFP fluorescence intensity is weak, and the differences is not so oblivious. It is recommended to extend the infection time or increase the viral infection MOI to enhance GFP intensity.

Reviewer #3: 1. Can the authors demonstrate which lysine residue of IRF3 undergoes deubiquitination by USP5?

2. Commercial antibodies for K48- and K63-linked polyubiquitination are available. It is good for the authors to detect endogenous IRF3 and its endogenous K48 and K63 ubiquitination using these antibodies?

**Part III – Minor Issues: Editorial and Data Presentation Modifications**

Reviewer #1: 1. The use of the word “restricts” in the title could be replaced by “inhibits” or “decreases.” The word restrict is usually used in this field in relation to restriction of virus infections (as opposed to restriction of innate immunity pathways that restrict virus infection as it is used here).

2. All bar graphs need to be re-made with symbols/colors that make more sense at first glance. The Key is confusing as open symbols are paired with solid bars and vice versa, making interpretation difficult.

3. The examination of STING KO cells in Fig 3 seems out of place and would not be a logical first place to start in terms of mechanistic examinations. This should be moved to later in the manuscript as a negative control or should be placed in the supplement.

4. Lines 180-190 describe the opposite of what the data show, and the conclusion at the end of this paragraph is the opposite of what they state in the previous lines. These are major mistakes in writing that are in opposition to the overall conclusions of the manuscript.

5. Line 192: This may be better written as “To explore roles of USP5 in RIG-I signaling, we assessed roles of…”

6. In Figure 5G, aren’t all of the ubiquitin chains shown in these blots expected to be anchored to IRF3 since it’s IP’d? It is thus confusing that the authors conclude that USP5 removes certain unanchored chains. Please clarify/explain further.

-Jacob Yount, The Ohio State University

Reviewer #2: 1.The manuscript contains several grammatical errors and types. The authors should check the manuscript carefully by themselves or use professional editing services for assistance.

2.In line 33, 'IFNAR' is mentioned for the first time, so its full name should be added.

3.In lines 149-151, to enhance readability, the text could be compressed by not repeating details and moving technical issues to the Materials and methods section or Figure legends.

4.Please provide the information in the methods section regarding the concentration of MG132.

5.Please discuss the potential role of unanchored ubiquitin in regulating host antiviral immunity against DNA viruses, since the cGAS-STING signaling pathway also requires IRF3 for induction of type I interferon.

Reviewer #3: 1.Figure 1F (left): the authors reported that viral infection can downregulate USP5 protein levels. However, in WT group, VSV upregulated Usp5 levels at time point 4 hrs. Please explain. In addition, total STAT1 levels were missing in Figure 1F.

2.Figure 2C, 2D: does USP5 overexpression or knockout affect virus titers?

3.Figure 3A-3C: does USP5 overexpression or knockout regulate basal expression levels of IFNB and ISGs?

4.Figure 3E: the p-STAT1 bands in Figure 3E are quite different from those in Figure 3D. Why? I would suggest to repeat Figure 3E to get clearer data.

5.Figure 4D: it is difficult to observe the staining of both USP5 and IRF3 in Figure 4D.

6.The authors reported that RNA viruses downregulate USP5 expression and that USP5 promotes RNA virus infection. How does USP5 affect DNA viruses?

7.In Figure 3F, the number of cells in each group varies considerably. Does USP5 regulate cell proliferation?

8.The description of results from lines 183 to 188 contains inaccuracies.

PLOS authors have the option to publish the peer review history of their article (what does this mean?). If published, this will include your full peer review and any attached files.

Reviewer #1: No

Reviewer #2: No

Reviewer #3: No
---

## [Decision Letter · Decision Letter 1]

17 Dec 2024

Dear Professor Ma,

We are pleased to inform you that your manuscript 'USP5 inhibits anti-RNA viral innate immunity by deconjugating K48-linked unanchored and K63-linked anchored ubiquitin on IRF3' has been provisionally accepted for publication in PLOS Pathogens.

Best regards,

Andrew Mehle

Academic Editor

PLOS Pathogens

Benhur Lee

Section Editor

PLOS Pathogens

Sumita Bhaduri-McIntosh

Editor-in-Chief

PLOS Pathogens

orcid.org/0000-0003-2946-9497

Michael Malim

Editor-in-Chief

PLOS Pathogens

orcid.org/0000-0002-7699-2064

Reviewer Comments (if any, and for reference):

Reviewer's Responses to Questions

**Part I - Summary**

Reviewer #2: The authors have addressed all my conerns. No further comments.

Reviewer #3: (No Response)

**Part II – Major Issues: Key Experiments Required for Acceptance**

Reviewer #2: (No Response)

Reviewer #3: All concerns I raised have been well addressed.

**Part III – Minor Issues: Editorial and Data Presentation Modifications**

Reviewer #2: (No Response)

Reviewer #3: (No Response)

PLOS authors have the option to publish the peer review history of their article (what does this mean?). If published, this will include your full peer review and any attached files.

Reviewer #2: No

Reviewer #3: **Yes: **Hui Zheng

---

## [Editor Report · Acceptance letter]

25 Dec 2024

Dear Professor Ma,

We are delighted to inform you that your manuscript, "USP5 inhibits anti-RNA viral innate immunity by deconjugating K48-linked unanchored and K63-linked anchored ubiquitin on IRF3," has been formally accepted for publication in PLOS Pathogens.

Best regards,

Sumita Bhaduri-McIntosh

Editor-in-Chief

PLOS Pathogens

orcid.org/0000-0003-2946-9497

Michael Malim

Editor-in-Chief

PLOS Pathogens

orcid.org/0000-0002-7699-2064